# Current Trend and New Opportunities for Multifunctional Bio-Scaffold Fabrication via High-Pressure Foaming

**DOI:** 10.3390/jfb14090480

**Published:** 2023-09-19

**Authors:** María Alejandra Fanovich, Ernesto Di Maio, Aurelio Salerno

**Affiliations:** 1Institute of Materials Science and Technology (INTEMA), National University of Mar del Plata, National Research Council (CONICET), Mar del Plata 7600, Argentina; mafanovi@fi.mdp.edu.ar; 2Department of Chemical, Materials and Industrial Production Engineering, University of Naples Federico II, 80125 Naples, Italy; asalerno@unina.it

**Keywords:** scaffold, gas foaming, tissue engineering, polymer, bioactivation

## Abstract

Biocompatible and biodegradable foams prepared using the high-pressure foaming technique have been widely investigated in recent decades as porous scaffolds for in vitro and in vivo tissue growth. In fact, the foaming process can operate at low temperatures to load bioactive molecules and cells within the pores of the scaffold, while the density and pore architecture, and, hence, properties of the scaffold, can be finely modulated by the proper selection of materials and processing conditions. Most importantly, the high-pressure foaming of polymers is an ideal choice to limit and/or avoid the use of cytotoxic and tissue-toxic compounds during scaffold preparation. The aim of this review is to provide the reader with the state of the art and current trend in the high-pressure foaming of biomedical polymers and composites towards the design and fabrication of multifunctional scaffolds for tissue engineering. This manuscript describes the application of the gas foaming process for bio-scaffold design and fabrication and highlights some of the most interesting results on: (1) the engineering of porous scaffolds featuring biomimetic porosity to guide cell behavior and to mimic the hierarchical architecture of complex tissues, such as bone; (2) the bioactivation of the scaffolds through the incorporation of inorganic fillers and drugs.

## 1. Introduction

Advanced tissue engineering (TE) therapies for the repair of critical-size tissue defects, such as bone and osteochondral tissue, require three-dimensional (3D) porous scaffolds. These scaffolds act as an analogue of the extracellular matrix (ECM), providing all the necessary cues for cell growth and stimulating new tissue morphogenesis. These include size and shape to fit the patient’s specific defect, porosity, and pore architecture to allow for cell migration and 3D tissue ingrowth as well as spatial and temporal control of biological signals to stimulate the capacity of the body to regenerate itself after in vivo implantation (Figure 1). Great efforts have been made in the past few decades to achieve this ambitious goal, and, today, there is a wide library of biomedical materials, bioactive molecules, and processing techniques that can be integrated to build multifunctional ECM-mimicking scaffolds. This review focusses on the gas foaming (GF) technique. Since its first implementation for scaffold fabrication, back in 1991, the GF technique has gained increasing importance in the biomedical field and scaffold fabrication. Although computer-aided (CAD) fabrication techniques have revolutionized TE approaches, GF-based processes still provide technological and design features that make their use extremely powerful in scaffold design and fabrication. Most notably, nowadays, the combination of GF and CAD processes is used for advanced scaffold design and manufacturing.

The aim of this review is to provide the reader with the state-of-the-art GF for scaffold fabrication and to point out the way in which it is possible to control scaffold microarchitectural properties and bioactivity. This work starts with a description of the basic principles of the GF process, mainly high-pressure foaming, to elucidate the role of the different processing parameters in the control of the morphological and architectural properties of foams. Furthermore, this review describes the historical evolution of the GF-based process in the design and fabrication of TE scaffolds, from the beginning up to the last decade, to allow for a better understanding of the progress achieved in this research field. This includes approaches enabling one to design biomimetic porosity gradients within scaffolds, as well as scaffold bioactivation strategies to guide the cellular processes involved in the development of new tissue.

## 2. Basic Aspects of GF through the Pressure Quench Method

GF methods can be classified based on the way the blowing agent develops into the polymeric matrix into physical foaming and chemical foaming [1]. In TE applications, the use of chemical foaming, such as those processes using carbonate materials, is restricted as the residues developed during the chemical reaction remain inside the polymeric matrix and may affect scaffold biocompatibility. Therefore, foamed scaffolds for biomedical applications have been mainly fabricated via physical foaming. This process is based on the sorption of blowing agents, such as CO_2_ and N_2_, at high pressure within the biocompatible polymeric matrix, followed by the quench of the pressure to ambient and the development of the pores. In this section, we focus our attention on relevant aspects in high-pressure polymer foaming, and how it is possible to control the scaffold pore structure.

The GF process belongs to the so-called thermodynamically based techniques, since the mechanisms by which the porosity develops are based on the sudden variation in the thermodynamic equilibrium of the system. We can divide GF into three basic steps (Figure 1): (i) sorption of the blowing agent at high pressure and polymer swelling; (ii) release of pressure, system supersaturation, nucleation, and growth; (iii) stabilization of the porous structure. From a technological point of view, steps (ii) and (iii) can be regarded as a single step, as bubble growth and structure stabilization spontaneously progress to a complete end after vitrification or crystallization of the polymer [2]. After depressurization, the gas solubility limit is met, and the gas phase separates from the polymer phase. The pores generated upon the separation between gas and polymer phases tend to form a spherical shape to reduce interfacial energy, while further impingement of neighboring cells formed polyhedral pores and possibly the rupture of the cell wall and opening of pores [3].

Among the different chemicals, carbon dioxide (CO_2_), nitrogen (N_2_), and mixtures of these two elements are the blowing agents most used for the fabrication of scaffolds by GF [4,5,6,7,8]. In fact, CO_2_ and N_2_ are non-toxic, eco-friendly, inexpensive, and chemically inert [5,7]. Most notably, GF with supercritical CO_2_ (scCO_2_) is more advantageous for scaffold fabrication and bioactivation and has been widely investigated in the past few decades by several groups [4,6,7,8]. In fact, the combination of gas-like viscosity and liquid-like density of scCO_2_ results in simultaneous high diffusion rates and solvent power. As a direct consequence, scCO_2_ sorption/desorption affects the free volume and chain mobility of polymers and, ultimately, modulates key properties, such as viscosity, glass transition, melting points, and crystallization rate [4,6,7,8,9,10]. When scCO_2_ is used as the pore-forming agent, it does not leave residues because it reverts to gas after depressurization.

Classical nucleation theory may allow one to explain the effect of GF parameters on the nucleation rate of gas bubbles and then pores within the polymeric matrix [1,11]. This theory correlates the final properties of a foamed material with the concentration and diffusivity of the blowing agent, the tension at the interface between the polymer/blowing agent and the nucleated pores, the temperature of the system, and the pressure drop [1,11]. Consistent with the nucleation theory, several studies showed that the pore size of the scaffold decreased with increasing pressure drop and with decreasing foaming temperature [4,7,9]. It is worth noting that during blowing agent sorption–desorption steps, polymers often undergo plasticization/vitrification and/or melting/crystallization processes. These transitions affect the viscoelastic properties of the polymeric matrix and, therefore, influence the morphology of the scaffolds and the pore architecture [12,13]. For example, using a batch foaming technique, Yang et al. showed that polylactic acid (PLA) crystallization during the sorption of scCO_2_ hindered the foaming of the sample and that a higher foaming temperature (120 °C) is required to obtain interconnected pore structure [13]. Most notably, the architecture of the scaffolds also depends on the temperature and pressure profiles during both the saturation and foaming steps. For scaffolds fabricated through scCO_2_ foaming, faster depressurization produced more homogeneous pore distributions and smaller pores. In contrast, the decrease in the depressurization rate resulted in scaffolds with larger pore size distributions and larger and more interconnected pores [14,15,16]. The explanation of this effect is that the pore nucleation time period is affected by the time period over which the thermodynamic instability is induced in the system. In fact, at each time step during depressurization, a shorter depressurization time (e.g., higher depressurization rate) means a higher pressure drop, and, therefore, more pores nucleate within the polymeric matrix (the number of cells nucleated increases exponentially) [17,18]. On the contrary, by increasing the depressurization time, the time required for nucleation is longer, and, therefore, some of the blowing agent dissolved in the polymeric matrix diffuses into the pores to promote its growth [17,18]. The effect of depressurization time is complicated further by the fact that faster depressurization accelerates the cooling of the system and, therefore, the vitrification/crystallization of the polymeric matrix [19,20].

In the next section, we describe the evolution of GF in the design and fabrication of TE scaffolds, highlighting the most important advancement of this technique in the biomedical field.

## 3. Overview of Polymeric Scaffold Fabrication via GF-Based Processes

As shown in Figure 2, our description of the evolution of GF-based processes for the development of TE scaffolds considers the main challenges achieved over time in this field by dividing the research articles into three main groups. The first group includes works published from 1991 to 2005, as they represented the first attempt to fabricate porous biodegradable foamed polymers and composites for the development of TE scaffolds. In the second group, the decade 2006–2015, the scientific community’s attention was mainly focused on optimizing the composition and structure of gas foam scaffolds to meet the morphological, biomechanical, and biochemical requirements of different native tissues. In the third group, from 2016 to the present, advanced foaming strategies, such as those based on the combination of GF and CAD technologies, are presented and critically discussed in Section 4 and Section 5 of this review.

GF for scaffold preparation was introduced for the first time in 1991 in the patent of De Ponti et al. The authors described the fabrication of porous biodegradable foams made of polyesters, such as PLA and polylactic-co-glycolic acid (PLGA), via scCO_2_ foaming for pharmaceutical applications, such as surgical implantation or controlled-release drug delivery systems [21]. Five years later, Mooney and co-workers published the first research article describing the use of compressed CO_2_ foaming to obtain porous polyester scaffolds for TE applications, avoiding the use of potentially harmful organic solvents for cells and tissue [22]. The GF process was successfully applied to PLA and its copolymers (PLGAs) with polyglycolic acid (PGA). The processing conditions were as follows: saturation at 5.5 MPa pressure, 20–23 °C temperature for 72 h followed by depressurization lasting 15 s. The scaffolds obtained had an overall porosity of up to 93% and an almost uniform distribution of macropores, 100–500 µm, throughout the polymer matrices. However, the pores were poorly interconnected, and the surface of the samples was characterized by a non-porous skin layer due to rapid CO_2_ diffusion during depressurization [22]. No foaming was observed for neat PGA material under the operating conditions tested. To increase the interconnectivity of pores both on the scaffold surface and inside, the GF process was combined with the particle leaching technique (PL) [23,24]. The PLGA particles were mixed with NaCl particles, sieved to achieve three different size ranges, with NaCl/PLGA weight ratio of 0 to 50, and compressed at room temperature to produce a solid disc. The discs were then loaded into a high-pressure vessel and exposed to CO_2_ gas at room temperature, 5.5 MPa pressure for 48 h to saturate the polymer with gas, followed by depressurization and polymer foaming. Subsequently, the NaCl particles were removed from the matrices by soaking them in water for 48 h. Scaffolds with interconnected porosity and open porous surfaces were achieved. The overall porosity of the scaffold was controlled by the NaCl/PLGA ratio up to a value of 97% and the pore size by the size of the NaCl particles. Furthermore, compared to scaffolds prepared by combining solvent casting and particulate leaching (PL) casting, GF/PL scaffolds exhibited more uniform pore structure and enhanced mechanical properties. The authors also studied the effect of the size of the particles in the starting NaCl/PLGA mixture on the morphology and interconnectivity of the scaffolds [24]. The results showed that, compared to large-particle-size scaffolds (250–425 µm), those prepared from smaller particle sizes (75 µm) provided more homogeneous porosity and greater pore interconnectivity [24]. Another potential advantage of the GF/PL process, compared to processing techniques using organic solvents, is that this process is likely to lead to a lower denaturation of growth factors incorporated within the matrix [24]. Vascular endothelial growth factor (VEGF) or plasmids encoding platelet-derived growth factor (PDGF) were mixed with PLGA and NaCl particles before foaming to prepare drug delivery porous scaffolds [25,26]. These scaffolds showed growth factor release over a period of days to months and increased ECM deposition and blood vessel formation after in vivo implantation [25,26]. The use of sucrose particles instead of NaCl particles as solid porogens for GF/PL PLGA scaffolds improved the encapsulation and release control of poly(ethylenimine) (PEI)-condensed DNA to optimize cell transfection in vivo [27,28]. This is because the ionic interactions that bind PEI to DNA can be disrupted at a high sodium chloride concentration, leading to the rapid dissolution of naked DNA from a PEI DNA scaffold [27]. The combination of GF and PL increased the open porosity of the scaffolds, but, conversely, made the fabrication process more complex. This is because the particulate porogen must be mixed with the polymeric matrix and further leached out by soaking the sample in proper solvent. This step is often quite long and may result in incomplete porogen removal and the undesired leaching of encapsulated drugs. To overcome these limitations, several studies investigated the effect of GF conditions on the fabrication and optimization of the pore structure of porous scaffolds. Singh and co-workers studied the rate of CO_2_ uptake and the equilibrium concentration of CO_2_ in PLGA as a function of saturation temperature, equal to 25 or 35 °C, and pressure, in the 5 to 50 MPa range [29]. Foaming experiments were carried out by saturating PLGA samples with CO_2_ at 10, 14, 15, and 20 MPa and at a temperature of 35 or 40 °C. Porous scaffolds with relative densities ranging from 0.107 to 0.232, overall porosity as high as 89%, pore size from 30 to 100 µm, and variable pore interconnection were obtained [29]. Compared to the PL scaffold, the PLGA scaffold prepared via scCO_2_ foaming released a more basic fibroblast growth factor (bFGF) per gram of polymer, while producing a slower release rate of the active factor, suggesting a possible deactivation effect of scCO_2_ [30]. Biodegradable polyesters typically have glass transition temperatures in the range of 40–50 °C and, therefore, undergo a vitrification/plasticization transition when solubilized with CO_2_, under subcritical or supercritical conditions, at temperatures below or equal to 40 °C. Higher temperatures were required for CO_2_ foaming of polymers, such as PLA, characterized by high molecular weight and crystallinity degree [31,32]. For bone TE applications, PLA was first mixed with ceramic particles, either hydroxyapatite (HA) or β-tricalcium phosphate (β-TCP), to mimic the composition and structure of the mineral phase of bone [31,32]. The composite materials were then saturated with scCO_2_ at 195 °C and pressures in the 14.5–22 MPa range for 10 min. Foaming was optimized by assessing the effect of depressurization rate, in the range of 0.17 to 1.19 MPa/s, and simultaneously cooling the system temperature at 4.5 °C/s [31]. A wide range of morphologies, including open and closed pores, porosity, and compression moduli, were achieved with neat PLA and composite foams, with pore sizes ranging from 200 to 1000 µm. Furthermore, the addition of 5 wt% ceramic particles to PLA improved the elastic compression modulus of the scaffold and enhanced the expression of osteoblastic genes in vitro of human primary osteoblasts and human fetal bone cells [31,32].

Due to the growing need for multifunctional scaffolds for TE applications and following the previous success of the GF process for the development of porous scaffolds, the 2006–2015 decade witnessed the significant advancement in this field of research (Figure 2). Some works investigated the effect of the chemical composition of polymers, the molecular weight, and the processing parameters on the pore structure characteristics of porous scaffolds prepared via low- and high-temperature scCO_2_ foaming [7,33,34,35]. Tai and coworkers studied the low-temperature scCO_2_ foaming of a series of amorphous PLA and PLGA polymers with different inherent viscosities and compositions [7]. The processing parameters under investigation were saturation/foaming temperatures from 5 to 55 °C and pressure from 6 to 23 MPa. In agreement with previous observation, scaffold porosity and pore size were affected by CO_2_ sorption (e.g., temperature–pressure–time combination) and diffusion (e.g., foaming temperature and depressurization rate). The pore size of the scaffolds also decreased with increasing glycolic acid content in PLGA copolymers due to reduced CO_2_ solubility [7]. The application of time-lapsed imaging and image processing during scCO_2_ foaming allowed differences in scaffold plasticization times and the foaming process to be observed and optimized [33]. Lemon and co-workers proposed a computer algorithm applied to image data sets obtained from µCT analysis to quantify the interconnectivity (e.g., the fraction of open pores as a function of pore throat size) of scaffolds prepared by GF [34]. Ultrasonic pulse echo reflectometry was also proposed to noninvasively monitor the fabrication of the scCO_2_ foaming scaffold online and to correlate the results obtained with those achieved using µCT analysis [35]. Low-temperature scCO_2_ foaming was also proposed for the rapid production of biodegradable PLA scaffolds containing mammalian cells in a single step [36]. Using optimal cell survival conditions, namely 7.4 MPa, 35 °C, and up to 3 min of saturation, scCO_2_ was used to process a mixture of PLA and cell suspension, and, upon pressure release, a polymer scaffold containing viable mammalian cells was formed.

Semicrystalline polycaprolactone (PCL) scaffolds were also manufactured using low-temperature scCO_2_ foaming [10,37,38,39,40]. The solubility and diffusivity of CO_2_ in PCL are influenced by both the molecular structure and the crystallinity of the polymer [37,38]. Furthermore, as previously commented, different crystallinity can result in porous scaffolds with different porosity and pore architecture [10]. The melting point depression of semicrystalline polymers, such as PCL, exposed to compressed CO_2_ was investigated as a function of applied pressure and temperature using techniques, such as infrared spectroscopy, light transmission analysis, capillary method, and shear viscosity measurement [38,41,42,43,44]. According to the reported data, at relatively low CO_2_ pressures (0.5–1 MPa), the melting point of PCL first slightly increases and then, after reaching a maximum, it starts to decrease as the CO_2_ pressure further increases, down to a value of 35 °C approximately at 10 MPa. A minor decrease in the melting point of the polymer was observed with increasing CO_2_ pressure, as the mobility of the chain is balanced by the hydrostatic pressure, and the melting point remains constant over a wider range of pressure. These considerations were also corroborated by visual observation of PCL samples under CO_2_ pressure [37,40]. However, in the case of semicrystalline copolymers of ε-caprolactone and x-pentadecalactone, porous scaffolds have been reported to develop only when scCO_2_ saturation was carried out at a temperature higher than polymer melting [45].

The incorporation of an inorganic filler within polymeric biomaterials is a useful way not only to improve the biomechanical and biological properties of scaffolds but also to improve foam control [46,47,48]. For instance, it was observed that the incorporation of an increasing amount of silica particles into the PLA scaffold not only reduced the pore size and increased the number of pores that were formed (nucleating effect) but also reduced the pore wall thickness and improved the opening of the pore wall. However, the effect of fillers on pore size and interconnectivity in scaffolds obtained by GF is still discussed. The addition of a filler also increases the viscosity of the polymeric matrix, and this effect may limit pore growth and favor closed pores, therefore, decreasing porosity, specific surface, and increasing the thickness of the pore wall [46]. Post-processing of polymeric foam scaffolds with ultrasounds was proposed to further improve pore connectivity and fluid transport [49,50]. Even when applied to thick PLA samples (up to 8 mm), the process led to an increase in the mean pore size, by approximately 10–20%, and pore interconnectivity without loss of scaffold integrity [50]. As a direct consequence, the fluid transport was enhanced, and it was possible to achieve 100% filling of the scaffold pore with water (over a timescale of a few hours), overcoming the polymer hydrophobicity.

A common phenomenon in tissue engineering is the formation of a dense tissue layer on the surfaces of the scaffold, restricting cell/tissue penetration and fluid diffusion at a distance greater than 200 µm and resulting in a necrotic core. Several studies addressed this problem by designing bimodal porous foam scaffolds with aligned pores throughout the scaffold structure [51,52,53,54]. Enhanced cell and tissue penetration was observed, both in vitro and in vivo, after the incorporation of 400 µm diameter aligned channels within the porous structure of the foamed PLA scaffold [51]. Similar results were observed for porous PCL scaffolds with a bimodal pore size distribution and prepared by combining GF and selective polymer extraction from co-continuous PCL/gelatin blends [52,53,54]. When the blends were foamed at a temperature lower than the melting of PCL (44 °C), scaffolds with small (40 µm) rounded pores (40 µm) and large (300 µm) tubular pores (300 µm) were obtained after gelatin leaching. Compared to monomodal PCL scaffolds that have a similar porosity value (60% ca.) and prepared via foaming at a temperature higher than the melting of PCL, bimodal scaffolds improved stem cell colonization, proliferation, and osteogenic differentiation within the entire 3D porous architecture [53,54]. The design and fabrication of scaffolds with multiscale porosity were also possible through two-step depressurization during scCO_2_ foaming [10]. In this process, PCL samples were solubilized at 37 °C and 20 MPa for 1.5 h and, subsequently, the pressure was released to an intermediate pressure of 9 MPa with a high depressurization time to allow for the formation of large pores. After equilibration of the system temperature to 37 °C, the pressure was finally quenched to the ambient temperature very quickly to induce nucleation of smaller pores and growth of existing ones [10]. Each of the two approaches described above has pros and cons. The use of solid porogens, such as NaCl particles or immiscible polymers (polyethylene oxide, gelatin), may allow for the almost independent control of overall porosity, pores size, and shape, which is difficult to achieve with the GF technique alone. On the contrary, the double depressurization approach does not require the two additional steps of blending and porogen leaching, which increase production time and may be detrimental to the possible incorporation of bioactive molecules.

We have previously highlighted that the low-temperature foaming of highly crystalline polymers, such as PLA and PCL, may be impaired by the blowing agent diffusion limitation within crystalline domains. The morphology of foamed scaffolds may depend on the randomly distributed nucleation of heterogeneous pores at the amorphous/crystalline interfaces. ScCO_2_ blowing agent mixtures with organic solvents, namely acetone or ethyl esters (ethyl lactate and ethyl acetate), were used instead of neat scCO_2_ to improve polymer plasticization and foaming [16,55,56,57,58]. This approach was extremely useful in reducing the viscosity of polymer/fill composites to enhance foaming [57] or for the extrusion of tubular porous conduits for TE applications [56]. Most notably, by dissolving proper drugs in miscible organic solvents with scCO_2_ and mixing these drug/solvent solutions with polymeric powder followed by the GF process, porous scaffolds with a controlled delivery of bioactive molecules were fabricated [59,60].

For in vivo applications, porous scaffolds must be implanted at the defect site where the tissue needs to be repaired. The scaffold degrades at a rate comparable to the new tissue growth rate, and during this healing process, the porous structure must withstand cell traction forces together with external applied forces to the proper healing of the wound. Therefore, the biomechanical characteristics and degradation properties of the foamed scaffold are key determinants and have been the subject of different studies [61,62,63]. Leung and co-workers fabricated porous PLGA scaffolds using the GF/PL process and studied the effects of PLGA composition and porogen concentration on the morphology and mechanical properties of the scaffolds [61,62]. The authors obtained scaffolds with porosity values in the 85 to 95% range and reported that the varying composition of PLGA did not significantly affect the pore size and static compression modulus of the overall sample [61]. However, although the modulus remained nearly identical for scaffolds with the same relative density, the strength of the material during compression varied due to the different pore morphologies of the samples. White and co-workers reported similar results for PLA scaffolds prepared using the scCO_2_ foaming process, since they observed that Young’s modulus of scaffolds increased at high depressurization rates due to the increased relative density of the foams [14]. Both the type and the amount of porogen were found to have a significant impact on PCL scaffold porosity and, consequently, on the mechanical behavior of the final scaffold [5,53,54]. The effect of scaffold degradation on the mechanical properties of PLGA scaffolds prepared via the GF/PL process was studied in [63]. During 90 days of soaking in different degradation media, namely dH_2_O, phosphate buffered saline, and cell culture medium, the scaffold retained its integrity, and the mechanical properties varied slightly. This effect was ascribed to the rearrangement of smaller polymeric chains during the degradation process that induced the shrinkage of the scaffolds and the decrease in porosity [63].

## 4. Advanced Control of Scaffold Architecture

We have seen that it is possible to fine-tune the morphology of the scaffold porosity by selecting the type and conditions of the process, exploiting the structural characteristics of the polymer. Optimal pore size, size distribution (single- or multi-modal), pore shape (isotropic or non-isotropic), and degree of interconnection (open or closed-celled), can be achieved *uniformly* throughout the scaffold. However, this (spatial) uniformity seems too artificial: when analyzing natural tissues, in fact, we realize that *graded* structures are much more abundant than uniform ones. In porous natural structures, such as human bones, all of the mentioned features of the pore morphology vary spatially, nonmonotonically, in different directions, forming an incredibly complex and, needless to say, optimized architecture (Figure 3).

Artificial scaffolds do not stand out in front of this complexity. More importantly, the opportunity to design the scaffold by including gradients is an obvious chance to improve functional and structural performance. Inducing morphology gradients in porous structures through gas foaming has to be achieved by controlling the different steps described in Section 2. In particular, the nucleation step, where bubbles are formed, is the key in this instance. Nucleation itself is a fast phenomenon, occurring in fractions of a second; furthermore, its comprehension is still limited and far from quantitative prediction. For the above reasons, manipulating the expanding matter prior to the pressure quench inducing nucleation appears to be a way to induce morphology gradients. To list the available methods to do so, the classical nucleation theory, CNT, can be invoked, although it is not quantitative in describing the bubble nucleation from polymer/gas solutions. In the framework of CNT, the stationary nucleation rate, *J_S_*, that is, the number density of stable nuclei forming per unit time, depends, among others, on the degree of supersaturation of the solution at pressure quench and on the physical properties of the polymer/gas system (for instance, the mutual diffusivity and the interfacial tension). Ubiquitous in the foaming practice, the presence of nucleating agents, i.e., additives that favor the bubble nucleation, may alter, typically enhancing, *J_S_*. This effect is typically addressed in the CNT as the transition from homogeneous to heterogeneous nucleation. In fact, the strategies observed in the literature that have proved effective in designing and inducing gradients in the foam morphology are as follows: (A) use of gradients in the density of the number of nucleating agents; (B) the inducement of temperature gradients; and (C) the inducement of gas concentration gradients (see Figure 4).

### 4.1. Nucleating Agent Gradients

Several ways have been reported to localize nucleating agents in expanding polymeric matter and, as a consequence, preferentially localize bubble formation. For instance, Yu et al. [63] prepared foams with graded cell size by compression molding a treated anodized aluminum oxide film onto several polymers, including polystyrene, polymethyl methacrylate, and polylactic acid. Capable of inducing enhanced nucleation, the film was responsible for attaining nanometric bubbles at the polymer/film interphase. This nucleation effect faded towards the bulk of the polymer, where micrometric bubbles were observed, thereby inducing the formation of a graded cellular structure. When the foaming conditions and polymer changed, morphological features, including cell size, cell density, and cell size gradient range, were nicely tuned. Of course, the surface effect spread over a limited thickness in the bulk of the foamed part and graded cellular morphology of about a few hundreds of microns could be achieved. To achieve the bulk effect, nucleating agents should be included in the volume of the foamed part. Doing so in a graded manner could be tricky. In this context, Pinto et al. found an interesting way to non-uniformly disperse nucleating agents into polymethyl methacrylate by exploiting the in situ synthesis of ZnO [64]. The synthesis of ZnO nanoparticles from Zn(OAc)_2_ was proposed under a temperature gradient. This led to a ZnO-Zn(OAc)_2_ spatial transition from the hot end of the part, where the Zn(OAc)_2_ -> ZnO synthesis occurred, with the formation of heterogeneous nucleation-efficient nanoparticles, to the cold end of the part, where Zn(OAc)_2_ remained unreacted. Standard batch foaming, i.e., at uniform temperature, with CO_2_ of the so-attained polymethyl methacrylate parts with nucleating agent gradients led to the formation of pores with sizes from 0.1 μm ca. in the nucleated part to 1 μm ca. in the non-nucleated part, over a span length of several tens of mm, which is compatible with scaffold size. As observed by these few examples, it is possible to achieve a graded foam by the use of concentration gradients of nucleating agents. However, the process does not seem to offer a multitude of design variables. Coupling with the methods described in the following sections can be a way to better tune the morphology to the application.

### 4.2. Temperature Gradients

Foaming temperature is the most effective among the processing variables (the other being the *type* and *concentration of blowing agent*, and the *pressure drop rate*, which sets the thermodynamic thrust bringing about bubble formation in the pressure quench foaming method, which is the most common). Imposing a temperature gradient in the expanding matter before foaming is then the most effective way to induce gradients. This is feasible considering the typical thermal diffusivities of polymers (*a*) in the order of 10^−7^ m^2^·s^−1^, and thermal conductivities (*k*) in the order of 10^−1^ W·m^−1^·K^−1^. This means that, when a surface is in contact with a heat source that imposes a certain temperature, the time, *τ*_energy_ (i.e., the characteristic time of the energy transport), required for the temperature change to be felt at a certain distance, *L*, is in the order of *τ*_energy_
*= L*^2^*/a*. In the case of a scaffold with a characteristic size, *L*, of 10 mm, *τ* is in the order of 10^3^ s, slow enough with respect to the foaming time, typically a fraction of a second. In stationary experiments, the low thermal conductivity allows one to impose a gradient with minimal energy consumption. The first evidence of foams with a graded morphology brought about by a temperature gradient is not easy to trace back in the literature because it is the most common (rarely desired) result in laboratory practice. In fact, the effect of the foaming temperature on the resulting foam is so strong that even small (say 0.3 K) temperature differences can be easily detected at foam inspection. The first experiment designed with the purpose of applying a temperature gradient and measuring its effect on the production of a porous material was provided in 2015 by Bai et al. [65] to freeze-dry a water-hydroxyapatite slurry. In 2016, Kiran and co-workers [66] introduced a gas foaming experimental set-up capable of imposing temperature gradients as large as 100K over a length of 25 cm, which allowed, in a single experiment, exploration of the entire processing temperature window for a given polymer. The reported cylindrical configuration of the pressure vessel could be easily modified and adapted to different scaffold geometries. Several studies adopted the approach reported in numerous thermoplastics, proving its versatility [67].

### 4.3. Gas Concentration Gradients

As anticipated in the preceding section, the gas (or blowing agent) concentration in the polymer prior to the pressure release is an important processing variable and can be widely utilized to define the foam features. The nature of the blowing agent (e.g., among a multitude of available chemicals, carbon dioxide or nitrogen) and the solubilized amount represent an effective tool to tune the foam density and pore morphology, as abundantly reported in the literature [68]. As such, any gradient of the blowing agent nature or amount solubilized in the polymer would give a graded foam. Before proceeding, it is worth noting that it is hard to conceive of a way to induce gradients by locally manipulating the pressure drop rate, the third foaming variable, as typically pressure is released in fractions of a second. Recently, Trofa et al. introduced a way to induce blowing agent concentration gradients within the expanding polymer by exploiting time-varying boundary conditions in blowing agent sorption prior to foaming [69]. In fact, typical blowing agent diffusivities in polymers are in the order of 10^−7–^10^−6^ m^2^·s^−1^, which calls for faster action with respect to the energy transport problem from the preceding section. Imposing a concentration gradient of blowing agents can be achieved by a quick change in the headspace of the blowing agent in the pressure vessel, with the characteristic mass transport time of mass transport of *τ*_mass_
*= L*^2^*/D*. In the case of *L* = 10 mm, *τ*_mass_ is in the order of 10^1^–10^2^ s, compatible with modern pumping systems. The authors reported numerous examples of multigraded foam morphologies that were possible with ingenuous pressure histories and provided a modelling tool to design the layering according to the needs. A preliminary experiment reported the use of the method to achieve complex shapes endowed with morphological gradients, such as a 1:10 scale human femur. The authors adopted polystyrene as a model polymer and polycaprolactone for the femur, showing the versatility of the method for any foamable biomedical thermoplastic [70,71,72,73,74]. Most recently, the gas foaming method with time-varying boundary conditions has been coupled with additive manufacturing, providing additional versatility in terms of porosity architecture [75]. This is valuable, especially in tissue engineering applications, as the use of combined techniques paves the way for porous heterostructures, which are structures composed of micro and macro pores, whose morphology can be tuned to a specific application through manufacturing and synthesis. Research in this field is mainly focused on multiple material structures, and the most diffused and studied porous heterostructures are layered structures containing an interphase between different materials. Focusing on a monomaterial heterostructure, the easy tunable morphology of porous heterostructures gives favorable mechanical, thermal, and acoustic properties [76].

Recently, the additive manufacturing of polymers has proven to be very effective in reducing manufacturing costs and improving design flexibility, enabling rapid multi-prototyping without using costly instruments such as injection molding [77,78,79]. Further, 3D printing technology has been widely used in biomedical fields for customization, rapid prototyping of complex structures, and low cost [80]. Among 3D-printing methods, fused deposition modeling (FDM) is a solvent-free method, in which the melted filament is extruded from the nozzle and then stacked layer by layer according to predefined patterns [81]. It is suitable for the processing of a wide variety of thermoplastics and for the realization of complex structures, both on a small and large scale [82]. The limit of FDM is the resolution, today ~100 µm, not comparable to gas foaming, when, nowadays, sub-micrometer pores are produced (nanocellular foams). Therefore, coupling FDM with gas foaming may provide3D-printed foamed structures with a complex geometry and both macro- and micro-porosities [75,83,84,85].

Today, the achievement of porous structures at different scales with thermoplastic polymers, adapted to specific functional and structural requirements, appears feasible by combining different processing techniques. In fact, 3D printers can produce almost any shape of macroscopic objects with macroscopic porosity. The gas foaming process remains the only available technique that delivers porosity at the micrometer and sub-micrometer scales with a variety of morphological features, such as the size of the pores, the number of pores, and their orientation by selecting the suitable blowing agent and processing conditions.

## 5. The Bioactivation of the Scaffolds through Incorporation of Bioactive Fillers and Drugs

### 5.1. Addition of Filler Inorganic Micro/Nanoparticles

The strategy of producing composite foams with polymers and inorganic fillers to improve their mechanical and biological response has been utilized more and more times. Different efforts are still being made to build scaffolds with sufficient mechanical resistance to be colonized by bone cells, allowing for bone regeneration. Knowing the structure and dynamics of bone tissue is the starting point for the development of synthetic substitutes with specific functionalities. Bioactivation of scaffolds through the incorporation of bioactive filler arises from knowing the details of the bone structure and its metabolism of formation and regeneration. Recognizing bone as a dynamic tissue is essential. Briefly, bone is a specialized connective tissue made up of by an ECM, with two components, one organic, which represents 30–35% being 95% collagen type I and the non-collagenous matrix, and the other inorganic, with 65–70% in the form of HA, a basic calcium phosphate. Bone tissue is metabolically very active and is under constant remodeling, replacing old bone with new bone. Bone remodeling is carried out by osteoclasts, which are the cells responsible for the destruction (resorption) of old tissue, and by osteoblasts, which synthesize new tissue. All this is under the direction of a system of hormonal signals that, in turn, are modulated by local bone factors that maintain a balance. Under normal conditions, a human being renews the entire bone mass every 10 years [86]. Alterations in bone remodeling may lead to pathological conditions, such as osteoporosis, osteogenesis imperfecta, cancer, and infections, among others.

Usually, osteoconductive and osteoinductive compounds, such as β-TCP, HA, and bioactive glass (BG), have been included in polymeric matrices using conventional methods and then foaming using scCO_2_. The addition of this type of inorganic filler promotes cell adhesion, improving the behavior of foams made with low-hydrophilic polymers. In addition, it gives the resulting material improvements in its mechanical performance. However, during material processing with scCO_2,_ this type of additive increases the viscosity of the plasticized polymer by CO_2_ under pressure. Furthermore, the filler content directly influences the degree of crystallinity that the polymeric matrix will reach after foaming.

Table 1 shows the latest published works in which porous PCL materials were obtained with the addition of an inorganic filler, such as HA or bioactive glass particles. L. Diaz-Gomez et al. developed porous composite scaffolds containing PCL, fibroin, and/or nano HA (nHA) particles using a simple, straightforward, and reproducible processing method. The originality of the work focusses on scaffold processing in scCO_2_ one step, without organic solvents and avoiding the waste of materials. These authors were able to confirm that PCL-Fibroin-nHA scaffolds improve MC3T3 cell attachment and proliferation and induce bone repair more efficiently than other scaffolds that also lack growth factors, from in vitro and in vivo evaluations, which results in a promising approach for bone tissue engineering and bone regeneration [87].

Moghadam et al. reported the fabrication of composite scaffolds based on HA distributed through PCL blends with two different molecular weights using scCO_2_ foaming. These authors focused on evaluating the effect of the blend composition (high molecular weight)/(low molecular weight), namely HPCL/LPCL ratio on pore size, morphology, and mechanical properties. Adding HA nanoparticles to PCL blends reinforced the mechanical properties, but the porosity and pore size were reduced. Higher concentrations of LPCL produce larger pore diameters and less uniform ones in HPCL/LPCL blend scaffolds, in which the optimal ratio was found at HPCL/LPCL 60/40. These authors obtained a composite of HPCL/LPCL 60/40 containing 2.5 wt% HA processed at 140 bar and 45 °C with good biological properties [88].

A more complex procedure to obtain PCL composite scaffolds with HA or halloysite nanotubes (HNTs) as an inorganic filler was published by X. Jing et al. [89]. Extrusion compounding of PCL and polyethylene oxide (PEO) blends and foaming processes was performed on a twin-screw extruder equipped with a supercritical N_2_ supply system. The porosity and pore interconnectivity of the foamed scaffolds improved through the leached-out PEO phase. In the composite scaffolds, PCL is used as the matrix material, HA or HNTs are used as fillers to alter the properties and biological activities of the scaffolds, and a water-soluble polymer, PEO, is used as the sacrificial phase to enhance the porosity and pore interconnectivity of the scaffolds, since the pores generated through extrusion foaming are mostly closed. It was found that at the same concentration, compared with HA, HNTs showed a higher improvement in the viscosity of the compound. The addition of fillers reduced the pore diameter, and the HNT scaffolds showed lower pore diameters than those of the HA scaffolds due to the higher viscosity and stronger nucleation effect caused by the high aspect ratio and smaller filler size. The compressive properties of the PCL/HNT scaffolds are higher than those of the PCL/HA scaffolds that have the same filler content. The results demonstrated cells viability on all scaffolds and that 5% HA and 1% HNTs play a significant role in regulating cell growth. This study shows that the HNT filler can be a good alternative to the use of HA particles [89].

Other works propose the fabrication of polymeric scaffolds filled with bioactive glass particles. Bioactive glasses are well known to act positively in angiogenesis processes as a result of the dissolution of ions from their structure. Similarly, its ability to chemically join natural bone has been reported since the composition of the first bioactive glass was known. C. Song et al. fabricated highly interconnected macroporous scaffolds from mesoporous bioactive glass particles (MBGs) and PLGA composite via scCO_2_ foaming method [90]. MBG/PLGA composites with different contents of MBG (0.5–20 wt%) were obtained through the combination of ultrasonic shake and mechanical stir. The strategy of these authors focusses on using high pressure and a long depressurization time. The increase in pressure is beneficial for generating interconnected structures via the enhanced plasticizing effects of CO_2_, and extended venting time favors the growth of pores to a large pore size by improving pore growth. The authors report that the incorporation of MBGs enhances the pore nucleation, resulting in a reduced porosity and pore size of the scaffolds. Therefore, to compensate for the negative effects of MBGs, the foaming temperature should be slightly increased. The addition of MBGs shows positive effects on biological response and an improvement in the strength and stiffness of the scaffolds [90].

Recently, a novel technique was used to fabricate composite scaffolds incorporating MBGs and bioactive lipids (Fingolimod, FTY720), which possessed synergistic cues of bioactive lipids and therapeutic ions to potently promote bone regeneration and vascularization. Incorporation was performed by coating the bioglass particles with fingolimod, a commercial drug. For this purpose, the scCO_2_ foaming technique was adopted to fabricate fingolimod-MBGs- PLGA composite scaffolds with appropriate mechanical and degradation properties as well as in vitro bioactivity. The scaffolds obtained enhance the formation of type H capillaries within the bone healing microenvironment to couple angiogenesis with osteogenesis to achieve satisfactory vascularized bone regeneration. These findings provide a promising strategy for bone regenerative medicine to develop efficiently vascularized engineering constructs to treat massive bone defects [91].

Other scaffolds containing MBG particles were developed by foaming with scCO_2_ from matrices of poly(propylene carbonate) (PPC) mixed with starch. Processing parameters were adjusted to optimize porosity (ranged 50–60%), pore size (from 100 to 500 μm), pore interconnectivity (>76%), and mechanical properties. Additionally, an improvement in mechanical properties and good biological behavior was achieved due to the presence of starch and bioglass microparticles [92].

Another material that has started to be included in composite scaffolds is graphene. There is still tremendous research going on all over the world on this exciting material and its compounds, due to its attractive properties [93]. In addition, various methods have been used to prepare graphene-based materials, even supercritical fluid (SCF) technology. Graphite exfoliation can be performed with scCO_2_ to obtain graphene, showing distinctive advantages [94]. Other structural forms, such as graphene oxide (GO) and reduced graphene (rGO), are used for their properties in composite materials. In particular, the properties of low-cost graphene nanoplates (GNPs), such as intrinsic transport and mechanical properties, are inferior to those of monolayer or few-layer graphene. GNPs in polystyrene-based foams were fabricated using scCO_2_-assisted microcellular foaming [95] with the aim of fabricating GNP-reinforced composites with a wide range of uses. Composite scaffolds of PCL/GO and PCL/rGO were obtained via sc CO_2_ by S. Evlashin et al. [96]. They estimated the maximum concentration of GO and rGO to be ~2 wt%, since more content led to the formation of non-homogenous scaffolds. Composite PCL/rGO foams demonstrated good flexibility and the ability to undergo 105 loading cycles. However, the PCL/GO composites did not show flexibility and were destroyed under external loading. Cell adhesion to the PCL/rGO scaffold was better than that to the PCL and PCL/GO scaffolds. The rGO used is a material obtained via laser reduction of graphene oxide that contains less than 3 wt% oxygen and has an average crystallite size of 90 nm [97].

Poly (butylene succinate) (PBS) is a biodegradable polymer that has also been studied when foamed with excellent potential for application in tissue engineering. However, its low melt strength and high crystallinity result in poor foaming ability. To improve the performance of this polymeric matrix in the foaming process, some approaches have been introduced. With this aim, cellulose nanocrystals (CNCs) were used as a reinforcing nanofiller to enhance the strength, hydrophilicity, and degradation rate of the composite scaffold of PBS/CNCs. A well-defined controllable bimodal open pore interconnected structure was successfully fabricated through the synergistic control of temperature variation and a two-step depressurization in a scCO_2_ foaming process [98]. The open pore structure was characterized by a large pore (~68.9 μm in diameter) and a small pore (~11.0 μm in diameter), with a high open porosity (~95.2%). The scaffolds exhibited good mechanical compressive properties (compressive strength of 2.76 MPa at 50% strain), hydrophilicity (water contact angle of 71.7 °C), and good biocompatibility and viability of cells.

### 5.2. Addition of Drugs to Polymeric Matrix

To impart different types of activity to polymeric scaffolds, the inclusion of an active substance through different processes has been proposed. Basically, impregnation or functionalization processes using scCO_2_ are governed by the difference in the solute distribution between the supercritical phase and the polymeric matrix, which can be quantified with the partition coefficient. The retention of the solute in the matrix will depend on the interactions that arise between them. The interaction between the polymer and the drug is a major criterion that favors the impregnation and influences the molecular state of the drug into the matrix. It is possible to distinguish two types of interactions: (1) Weak or no interactions: the drug is carried by scCO_2_ into the matrix during the impregnation process and is then trapped in the matrix during the depressurization step. The drug often recrystallizes into the polymer due to the poor affinity with the matrix. This mechanism is also referred to as “deposition”. (2) Strong interactions: when the polymer and drug present a good affinity due to interactions, such as Van der Waals or H-bonding. These interactions are the driving force of the process and result in a partitioning of the drug in favor of the polymer. When these types of interaction are present, high drug loads distributed at the molecular level in the matrix can be achieved.

On the other hand, the solubility of the drug in scCO_2_ is relevant for the impregnation process, since the generation of polymer/drug interactions depends on it (the availability of the drug in the environment). The incorporation of drugs or active molecules can be physical or chemical. In the first case, the solute molecules are mixed with the polymer matrix and, therefore, remain free to move and can be dissolved by scCO_2_ or by a liquid environment, while the second case implies immobilization of solutes via covalent bonding with specific functional groups in the polymeric structure. Thus, it is possible to infer that the hydrophilic/hydrophobic properties of the polymer have a high influence on drug loading, conditioned to the hydrophilic/hydrophobic nature of the drug.

In recent years, polymer impregnation/functionalization with scCO_2_ has basically been carried out using two types of methodology:

(1) Physically separated matrix and drug: in this case, the amount of loaded drug depends on the dissolution kinetics of the drug in scCO_2_ and on the diffusion of the drug in the polymeric matrix (Figure 5a).

(2) Previously obtained polymer/drug compound: the prior preparation of the compound allows for better control of the amount of drug in contact with the polymer, favoring impregnation when there is low affinity between the polymer/drug (Figure 5b).

Both methods are influenced by temperature, CO_2_ pressure, secondary phases, soaking time, presence of cosolvents, and depressurization rate, among the most relevant [99,100].

The opposite effects of temperature on drug loading can be observed. When the drug is separated in the matrix (methodology 1), the effect of T should favor the dissolution of the solute in CO_2_ to increase impregnation. Here, we must consider the characteristics of the solute and know the crossover point, while in the processing of previously prepared composites (methodology 2), a low solubility of the solute in CO_2_ would be beneficial to maintain the drug load included in the matrix.

Something similar happens with pressure. In the first case, the increase in pressure leads to an increase in drug loading under isothermal conditions due to the simultaneous increase in drug solubility, CO_2_ sorption, and polymer swelling. However, the decrease or constant value of drug loading with increasing pressure has also been observed and was justified by the authors with the CO_2_/drug interactions that tend to prevail over the polymer/drug interactions. Moreover, in the second method, the pressure must be adjusted to achieve a good foaming process and minimize drug solubility.

The effect of the rate of depressurization on the extent of impregnation is not truly clear. One review reported that if the drug has a poor affinity for the polymer, it can be easily vented with CO_2_ from the matrix. In that situation, a high depressurization rate favors the entrapment of the drug in the polymer. On the contrary, for a system with strong polymer/drug interactions, a slow depressurization rate should be chosen [99]. On the other hand, Machado et al. [100] recently reported that rapid depressurization generally induces excessive solute losses, especially if the drug–polymer affinity is low. In this sense, most of the reviewed works selected low decompression rates. Therefore, the process variables must be previously studied in order to adopt an efficient impregnation mechanism. Other approaches have also been implemented to enhance impregnation. A strategy to improve the foaming of PCL supports with a drug is the addition of polyethylene glycol (PEG) as a secondary phase. PEG was added as a plasticizer to improve the morphology of the PCL scaffold, producing scaffolds with an open and interconnected structure. According to Guastaferro et al. [101], PEG can act as a viscosity reducer and can enhance the ability of PCL to swell and dissolve in water. these authors added theophylline in the PCL/PEG matrix using DMSO solutions until obtaining a gel, which was then foamed. In this way, a prolonged release of the drug was obtained after its encapsulation into the PCL/PEG scaffolds, whose presence contributed to slowing THEO release in the surrounding liquid medium. Polymer erosion and drug diffusion were the controlling mechanisms of theophylline release (Table 1, Part B).

Also, Y.X.J. Ong et al. [102] prepared a uniform dispersion of the drug in the polymer matrix of PLGA by adding PEG. A two-step fabrication process was used that combined emulsification solvent evaporation methods for encapsulation of the drug in PLGA microparticles followed by supercritical gas foaming. The encapsulation and release of model hydrophobic drug (Curcumin) and model hydrophilic drug (Gentamicin) were investigated, showing the potential application of this methodology for a wide range of active ingredients. With this methodology, it is possible to minimize residual organic solvents in the final product due to the high affinity of scCO_2_ for them. This paper showed that the drug release profile can be engineered by the selection of different PLGA polymer blends, varying the lactic to glycolic ratio and the molecular chain length of the polymer, and by the addition of compatible biodegradable polymers, such as PEG, to the polymer matrix.

On the other hand, Elham Khodaverdi et al. [103] focused on developing a copolymer with greater hydrophobicity, using a central composite design. Thus, a PEG-PCL-PEG (PEGCL) scaffold was functionalized with dexamethasone (DXMT). The drug was added to the PEGCL powder by mixing in solid state (10%) and compressed, applying hydraulic pressure to obtain disc-shaped tablets. The samples were then processed with scCO_2_ under optimized conditions. The loading capacity of the scaffolds decreased slightly after scCO_2_ treatment (8.97 ± 1.03%), indicating the partial extraction of DXMT from the supercritical phase during the scCO_2_ process (the loading capacity of pre-scCO_2_ scaffold = 9.09 ± 1.12%). The cumulative in vitro DXMT release assay revealed that, post-scCO_2_ treatment, the scaffolds delivered an almost complete release (79.18 ± 1.39%) following the Higuchi model (diffusion) as the main model explaining the DXMT release mechanism. However, these triblock copolymer scaffolds obtained through scCO_2_ foaming did not show a good pore microstructure, as assessed via mean pore size and distribution.

Different processes with great efficiency were developed to functionalize PCL supports with drugs. Among them, we can mention the works [104,105] where the authors implemented an integrated process that allowed them to sterilize a PCL matrix scaffold and functionalize the scaffold with vancomycin. The sterilization process was carried out by adding hydrogen peroxide in the high-pressure CO_2_ reactor, without being in contact with the polymeric material [104]. The inclusion of vancomycin was carried out by manually mixing a solid mixture of PCL/5% vancomycin [105]. Sterile PCL scaffolds with morphological characteristics such as natural bone tissue were obtained, achieving logR-6 sterilization levels against dry spores based on a dynamic procedure. This approach resulted in H_2_O_2_-free scaffolds without requiring post-processing aeration steps. Furthermore, the PCL-vancomycin scaffolds presented a relevant release pattern for prophylaxis and treatment of infections in the grafted area and supported the attachment and growth of MSCs without inducing their differentiation to a specific cell line [105]. As a relevant conclusion, the authors declare that because of the mild processing temperatures required, the sterilization and manufacturing of polymeric scaffolds incorporating thermolabile compounds such as monoclonal antibodies might be feasible.

Furthermore, porous PCL patches were impregnated with nimesulide, a non-steroidal anti-inflammatory drug, which has good solubility in supercritical CO_2_ [106]. The foaming of PCL and its impregnation with nimesulide were carried out in a one-step procedure. Release analyses via UV-vis spectroscopy revealed that nimesulide release was significantly delayed.

An optimized drug loading process to develop PCL/drug composite scaffolds was published by Salerno et al. [60,107]. The drug loading process was achieved by dissolving the drugs in miscible organic solvents with scCO_2_ and by mixing these drug/solvent solutions with PCL powder. Porous PCL scaffolds containing three different drugs, namely 5-fluorouracil, nicotinamide, and triflusal, were achieved [60]. ScCO_2_ saturation and foaming conditions were optimized to create porosity within the samples and to allow for the elimination of organic solvents. The drug loading efficiency was reported to depend on both the initial composition of the solution and the solubility of the drug in scCO_2_. The loading efficiency of highly soluble drugs with scCO_2_ was improved by adding a proper amount of free drug inside the pressure vessel. The drug delivery study showed that the release profiles depended mainly on the composition of the scaffolds and the characteristics of the pore structure.

The impregnation and foaming process can be carried out in a single step [108]. When the conditions of pressure, temperature, depressurization time, or type of polymer used are adjusted, microcellular scaffolds can be obtained with desired characteristics. Moreover, it has been demonstrated that the use of polymeric solutions allows the impregnation process to be carried out under mild conditions. In this work, gemcitabine impregnation in PLGA foams from polymeric solutions of ethyl lactate was studied. The effects of the polymer lactide to glycolide ratio (50:50 or 75:25), pressure (120 or 200 bar), and temperature (25 or 40 °C) were studied for three initial drug ratios (175, 105, or 35 mg GEM/g PLGA). The cell size of the foams varied between 35 μm and 158 μm, achieving an impregnation efficiency higher than 90%. Finally, a study of the release profile of gemcitabine in phosphate-buffered saline was investigated, and mathematical modeling was carried out. In this model, it was considered that the release process was divided into three different steps, controlled by the external diffusion in the first place, by the internal transfer of mass in the second, and then by the degradation of the polymer.

The single-step static scCO_2_ process employed (pressure of 30 MPa and temperature of 100 °C for 2 h) allowed for the fabrication of solvent-free polymeric foams and solid carvedilol dispersions with controlled microstructure and average pore diameter of 101–257 μm, suitable for application in the pharmaceutical industry [109]. ScCO_2_ did not remain in the foams after processing or affect the polymer composition, while carvedilol formed hydrogen bonds with the polymers. Carvedilol was molecularly dispersed in fabricated solid dispersions, and its transition from the crystalline to amorphous form was complete. The Korsmeyer–Peppas model was successfully used for the mathematical description of carvedilol dissolution from solid dispersions.

Polymeric matrices have also been functionalized with natural extracts with pharmacological properties [110,111]. Scaffolds were obtained through impregnation and foaming of PLA with thymol and thyme extract, a natural antibacterial agent, at 30 MPa and 100–110 °C. Thymol acts as a plasticizer, which results in an increased free volume of the polymer matrix of PLA and, consequently, higher gas sorption. This work clearly shows how the availability of the solute influences the wt% incorporated into the matrix. The authors obtained different degrees of loading of PLA with thyme extract through batch or integrated extraction and impregnation processes.

Salerno and Domingo reported on a batch foaming process based on solution scCO_2_ suitable for the preparation of polymeric foams with controllable composition, morphology, and pore structure [112]. The great advantages of the proposed approach for polymer foaming were also demonstrated by preparing composite PCL/TiO_2_ nanocomposite foams and PCL/5-fluorouracil foams with high filler and/or drug loading and large macropores in a single solution processing step. The results demonstrated that the proposed approach enabled the preparation of PCL foams with solvent residue close to 1 wt% of initial solvent weight and uniform porosity. The authors found that by increasing the affinity between organic solvents and scCO_2,_ the amount of solvent residue inside the foams can be greatly decreased to obtain foams with low density and small pores. Furthermore, as the polymer concentration in the starting solution increased, the viscosity increased, and the solvent residue and expansion ratio increased.

### 5.3. Simultaneous Addition of Filler and Drug

The reviewed publications report the fabrication of scaffolds by scCO_2_ foaming with a varied composition, both in terms of matrices and fillers. Part C of Table 1 shows reports developing porous structures from mixtures containing a PCL matrix, a filler, and a drug. These complex systems are strategically processed to achieve appropriate and reproducible drug delivery devices.

Highly porous PCL scaffolds containing biodegradable mesoporous microparticles (starch aerogel microspheres) and a bioactive compound (ketoprofen) were designed and produced by using impregnation/deposition and a foaming scCO_2_ process by Goimil et al. [113]. The use of starch aerogels as an admixture of the scaffold composition played a relevant role in the tuning of the ketoprofen release profile. The drug release falls into a complex diffusion-controlled mechanism, comprising a combination of diffusion through the matrix and through the pores filled with the solution from the release medium. Moreover, porous PCL scaffolds with thermosensitive enzymes entrapped were reported by G. Kravanja et al. [114]. They prepared a complex composite of PCL, chitosan (CS), HA, glutaraldehyde (GA), and enzyme transglutaminase (TGM) by mixing the components under supercritical conditions. Thus, the enzyme was cross-linked via GA to CS and changed its release patterns and preserved its activity.

On the other hand, PCL scaffolds loaded with different vancomycin contents (0 to 7 wt%) and CS were processed through supercritical foaming [105]. The addition of vancomicin was performed prior to foaming, through a solid mixture. Furthermore, the effect of the presence of CS, with antimicrobial properties, on the composition of the scaffold was evaluated. The mass transfer mechanism of vancomycin from the scaffolds was governed by the dissolution from two distinct fractions: the first fraction located on the outer surface of the large pores of the scaffolds, with a faster dissolution, and the second fraction located in the inner parts of the scaffolds, with a release from the PCL matrices governed by a complex interplaying diffusion and dissolution of the bioactive agent through the porous matrix. The presence of CS probably increased the vancomycin release rate due to the enhanced wettability of the scaffolds [105].

Li et al. [115] studied the addition of icariin (ICA) (main ingredient in *Herba epimedii*) as a bioactive with pharmacological effects in bone tissue engineering, such as angiogenesis, anti-osteoporosis, and anti-inflammatory. Along with ICA, Fe_2_O_3_ particles were incorporated into the PCL fibrous membrane via electrospinning, and then the 2D electrospun fibrous membrane was successfully expanded by depressurizing scCO_2_ fluid to obtain a 3D composite-layered fibrous scaffold. The highly porous magnetic 3D scaffold blending with bioactive ICA provided a new theranostic material with potential application in bone tissue engineering. The concept of theranostic scaffolding has been little developed to date.

**Table 1 jfb-14-00480-t001:** Polymer-based filler/drug products obtained by gas foaming-assisted impregnation/deposition.

Polymer Based Matrix	Filler or Active Compound/Drug	Foaming Conditions				Process Features	Observations/Evaluated Properties	Reference
		P	T	Soaking Time	dP/dt or Venting Time			
Part A: Matrix + inorganic filler								
PCL 50 KDa Fibroin 15–20 wt%	Nano hydroxyapatite (nHA) 10 wt%	140 bar	37 °C	1 h	3 bar CO_2_/min	One-step by scCO_2_ under stirring (500 rpm).	67–70% porosity Additives increased the compressive modulus, cellular adhesion and calcium deposition. synergistic effect of silk fibroin and nHA on the bone repair	L. Diaz-Gomez et al. [87]
PCL blends LPCL: 10,000 g/ Mol; HPCL: 70,000 g/mol	Nano Hydroxyapatite (nHA) 1–4 wt%	120–160 bar	45 °C	3 h	0.3 bar/s	100L, 60L, 60H, 100H and 60H-2.5%HA foamed by scCO_2_ in one step	Average pore size decreased from 612 μm to 132 μm and the porosity was reduced from 73% to 22.4% as the content of HPCL in the blends was changed from 0% to 100%. Optimal conditions: 45 °C and 140 bar.	M. Z. Moghadam et al. [88]
PCL 50 kDa PEO Mw:100.000	Hydroxyapatite (HA) (2 μm) 1, 5, 10 wt%		90–100			Blends PCL/PEO and composites were prepared by extrusion foamed using supercritical N_2_ (0.5%) (Screw speed 100 rad/min)	Highly porous (>75%). The HNT improved viscosity more significantly than HA, and reduced the pore size of scaffolds, while the mechanical performance of PCL/HNT scaffolds was higher than PCL/HA scaffolds with the same filler content. The cell differentiation for 5% HA and 1% HNT scaffolds were significantly higher than other scaffolds.	X. Jing et al. [89]
	Halloysite nanotubes (HNT) (800 nm) 1, 5, 10 wt%							
PLGA poly(lactic-co-glycolic acid) (mole ratio of LA:GA = 85:15, Mw = 50,000)	Mesoporous bioactive glass particles (MBGs) (5–20 wt%)	150–300 bar	38–50		from 2 min to 80 min	Bach foaming with previous sweeping the vessel with low-pressure CO_2_ for three times	Highly porous (73% to 85%). Highly interconnected (>90%). Pore size: 120 μm to 320 μm. MBGs reduce porosity, show positive effects on biological response, and improve strength and stiffness.	C. Song et al. [90]
PLGA (LA:GA = 85:15) Mw = 50 kDa	Mesoporous bioactive glass particles (MBG) 18 wt% Fingolimod-MBGs 18 wt%	250 bar	30–35 °C	1 h	80 min of venting time	Foaming from mixtures of polymer/MBG or polymer/FTY720-MBG	Developed scaffolds with angiogenic effects. Bioactive lipid and ionic products from the FTY/MBG-PLGA scaffolds synergistically improved vascularized bone regeneration.	S. Li et al. [91]
PPC Poly(propylene carbonate) 160 kDa Soluble starch powder ~25 μm	Synthesized bioglass microparticles	50, 75, 125	25, 30, 40	4 h	0.2, 2.5 and 10 bar CO_2_/s		Pore sizes: 100 to 400 μm (75 bar, 30 °C, 2.5 bar/s) Interconnectivity ~76%. Porosity 45–60% Enhancement in the mechanical behavior due to the presence of starch and bioglass microparticles.	I. Manavitehrani et al. [92]
PCL Mn = 80 kDa	Graphene oxide (GO) and reduced Graphene oxide (rGO). 0–2%	180 bar	80 °C	1 h-4 h	1 atm/s 100 atm/s	PCL/graphene prepared previously to foaming.	PCL/rGO foams with good flexibility. Cell adhesion to the PCL/rGO scaffold was better than that to the PCL and PCL/GO scaffolds.	S. Evlashin et al. [96,97]
PBS Poly (butylene succinate) (PBS,B601)	CNCs Carbon nanocellulose 0.5–5%	STEP1 22 MPa	110 °C	2 h	5, 10, 15 s respectively fast depressurization rate	synergistic control of temperature variation and two-step depressurization scCO_2_ foaming process	bimodal open-pore structure: large pores (~68.9 μm in diameter) and small pore (~11.0 μm in diameter). High open porosity (~95.2%). Compressive strength of 2.76 MPa, hydrophilicity (water contact angle of 71.7 °C)	J. Ju et al. [98]
		STEP 2 20, 18, 16 MPa	85 °C	10 min				
PCL Mn = 45 KDa	TiO_2_ (30%)	200 bar	50 °C	1–17 h	Two step depressurizations 1—venting 2—foaming	one-step process based scCO_2_ batch foaming	Pore-size: 200–1200 μm Low residual solvent High load efficiency	A.Salerno and C. Domingo [112]
Part B: Matrix + active compound/drug								
PCL Mn= 80,000 Da PEG(Mn= 10,000 Da), PEG/PCL ratio: 10–30 wt%.	Theophylline (THEO) 5 wt% to PCL	100–200 bar	40 °C	8 h		Foaming PCL/PEG/THEO gels from solutions in DMSO.	Optimization of gel matrix for foaming. The increase in PEG concentration led to an increase in the scaffold average pore diameter	M. Guastaferro et al. [101]
PLGA (Poly(D,L-lactic-co-glycolic acid) 75:25 Mw = 66–107 kDa PEG (10%)	Curcumin (CM) (~1% w/w) Gentamicin sulfate (GS) (~4 wt%)	120 bar	35	4 h	Not reported	Two-step process: 1-Drug-encapsulated PLGA powder (oil/water emulsion method) 2-CO_2_ foaming	Encapsulation Efficiency: GS: 25%, CM: 75% The release profile from all the samples suggests a diffusion-controlled model. No matrix degradation (2 weeks). Pore sizes: 55–120 μm.	Y.X.J. Ong et al. [102]
PEG-PCL-PEG (PEGCL) Mw= 13242.49 Da	Dexamethasone (DXMT)	234 bar	49 °C	2 h	5 min	Bach foaming	Maximum porosity %(79.18%) DXMT release by Higuchi model (Diffusion) (~79%)	E. Khodaverdi et al. [103]
PCL 50 kDa 1200 ppm of H_2_O_2_ 30% v/v/100-mL stainless steel reactor	5% Vancomycin hydrochloride (Mw 1486 g/mol, 94.3% purity)	140 bar	39 °C	2.5 h	5 g CO_2_/min	One-step by GF	Highly porous (>74%) 6-logarithmic reductions (logR-6) were reached for *B. atrophaeus, B. stearothermophilus and B. pumilus.* Vancomycin release profile was fitted to a bi-exponential drug release model.	V. Santos-Rosales et al. [104] C.A. García-González et al. [105]
L-PCL (10,000) H-PCL (80,000)	Nimesulide (NIME)	150–200 bar	35–40 °C	1–48 h	100 bar/min	one-step supercritical foaming + impregnation process	Solubility NIME varied from 0.035 mg/g CO_2_ at 60 °C/10.0 MPa to 0.55 mg/g CO_2_ at 60 °C/20.0 MPa. Maximum wt% NIME adsorbed: <1% for L-PCL, 35% for H-PCL. The release of NIME was delayed 3.5 times.	R. Campardelli et al. [106]
PCL 80 kDa	5-fluorouracil nicotinamide triflusal	200 bar	40 °C	1h	Two step	Venting and Foaming from compacted mixtures of PCL + drug solution.	Loading efficiency > 50% Pore size: 87–237 μm Optimized loading drug process.	A. Salerno et al. [60]
PCL Mn= 80,000 Da	5-fluorouracil 4.8% and 9.1%	200 bar	45 and 50 °C	1 h	(I) 0.03 MPa/s (7 MPa) (II) 0.1 MPa/s (3–4 min) or 2 MPa/s (5–10 s)	2-step foaming process from blends of 5-FU in DMSO mixed with PCL		A. Salerno et al. [107]
PLGA5050 (50 mol % lactic acid, 50 mol % glycolic acid), PLGA7525 (75 mol % lactic acid, 25 mol % glycolicacid)	Gemcitabine hydrochloride 105–175 mg/g PLGA	120–200 bar	25–40 °C	24 h		One-step process (impregnation and foaming) from 0.8 g PLGA/mL ethyl lactate solutions	Pore sizes: 35–158 μm, achieving an impregnation efficiency higher than 90%.	I. Álvarez et al. [108]
Soluplus^®^ (Mw: 90,000–140,000 g/mol) Eudragit^®^ (Mw: 150,000 g/mol) Hydroxypropyl methylcellulose acetate succinate (HPMC-AS) (Mw: 18,000 g/mol)	Carvedilol 30 wt%	300 bar	100 °C	2 h	1.5 MPa/min	Foaming from mixtures polymer/drug (mass ratio 1:0.3)	pore size: 75–560 μm (Soluplus^®^ and Eudragit^®^ foams); 90 to 340 μm (HPMC-AS foams) Carvedilol’s role as an additional plasticizer.	S. Milovanovic et al. [109]
Polylactide (PLA) Mw: 17 kg/mol polylactide-co-glycolide (PLGA-I, Mw: 95 kg/mol; PLGA-II, Mw: 17 kg/mol)	Thymol	75–150 bar	25–50 °C	2–24 h	0.5 MPa CO_2_/s	Impregnation and foaming from Polymer and drug separated by mesh	PLA foam, average pore diameter: 32.2–88.1 μm. Thymol loading of 0.92–6.62% (2–4 h). Concentration of thymol released within 5 h, 24 h and a month from the representative foams was in the range of 3.5–10 μg/mL, 5.5–15 μg/mL and 16.3–29.3 μg/mL, respectively.	S. Milovanovic et al. [110]
PLA-E Mw ≈ 210,000 g/mol PLA-I Mw ≈ 160,000 g/mol	Thymol Thyme extract	Extraction 300 bar	40 °C	2–5 h	3.8 ± 1.0 MPa/s.	-Batch process (BP) foaming and impregnation -Coupled extraction and impregnation process (CP)	Porosity > 65%. Pore sizes: ~15–200 μm. BP: thymol loading 4.7–8.5% and foam porosity ~60–65%. CP: 0.71–1.1% of thymol and 0.6–0.7% of thyme extract, porosity ~75%.	R. Kuska et al. [111]
		Impregnation 300 bar	110 °C					
PCL Mn = 45 KDa	TiO_2_ (30%) 5-fluorouracil (30%)	200 bar	50 °C	1–17 h	Two step depressurizations 1—venting 2—foaming	one-step process solution-based scCO_2_ batch foaming	Pore-size: 200–1200 μm Low residual solvent High impregnation/load efficiency	A. Salerno and C. Domingo [112]
Part C: Matrix + filler + active compound/drug								
PCL 50 kDa	starch aerogel microspheres (SAM) 1.2 μm Ketoprofen (KP) 0–5%	I STEP 150 bar	40 °C	11 h 1 h	0.9 g/min	(I) Impregnation of KP in SAM (II) Foaming PCL/SAM/KP physically mixed.	Porosity > 60%. Pore sizes: 75–99 μm. Interconnectivity > 75%.	L. Goimil et al. [113]
		II STEP 140 bar	37 °C		1.8 g/min	(II) Foaming PCL/SAM/KP physically mixed.		
PCL 80,000 g/mol	Chitosan (CS) 5–20 wt% Enzyme transglutaminase (TGM) glutaraldehyde (GA) Hydroxyapatite (HA) 5 wt%	150 bar	37 °C	6 h	0.05 MPa/s	(I) GF from blends of PCL + CS + HA + NaCl + TGM + GA mixed under supercritical conditions. (II) Salt leaching method.	Storage modulus 5,3–6 MPa. Mean pore size: 180–210 μm. Porosity 41–63%. composites with CS +5 wt% extend protein release patterns and preserved TGM activity up to one month.	G. Kravanja et al. [114]
PCL 50 kDA	Vancomicyn 0–7 wt% Chitosan 0–15 wt%	140 bar	40 °C	1 h	1.8 g/min	Powdered mixtures of PCL, vancomycin (V), and chitosan (Chit)	Porosity ~70%. Pore size: 20–90 μm.	C.A. García-González et al. [105]
PCL Mn = 80 kDa	Fe_2_O_3_ 0.24–1% Icariin (ICA) 0.1%					2D and 3D fibrous support by electrospinning and scCO_2_ foaming	Saturated magnetization of 2D fibrous membranes increased from 1.78 to 6.45 emu/g with 0.25% to 1% of Fe_2_O_3_, respectively. PCL/Fe_2_O_3_/ICA composites were expanded to 3D scaffolds after depressurization of sc CO_2_.	K. Li et al. [115]

## 6. Conclusions and Recommendations for Future Work

In the past few decades, the high-pressure foaming process has been widely used to fabricate 3D porous scaffolds for both in vitro cell growth studies and in vivo tissue regeneration. In fact, these processes allowed for the design of biocompatible and biodegradable scaffolds that feature important properties, namely a tailor-made architecture and bioactive features, and are manufactured using biosafe processing conditions for possible clinical use.

Scaffolds with gradients of pore size, density, and orientation are required to meet the biological and biomechanical requirements of musculoskeletal tissues, such as bone. In the case of foaming techniques, this issue was addressed by using gradients of nucleating agents, temperature, and gas concentration. In this context, the use of gradients of bioactive inorganic fillers in combination with low-temperature foaming techniques should be further explored to mimic the composition, architecture, and mechanical properties of composite structural tissues, like bone.

The optimization of different strategies to incorporate biomolecules, such as drugs and growth factors, is another important issue to enhance the regenerative properties of foamed scaffolds. Scaffold bioactivation was achieved by using the solvation properties of some blowing agents, such as CO_2_ in the supercritical state, which allowed for the homogeneous impregnation of drugs and biomolecules within several polymers. It is worth noting that processing biomolecules at high pressures may affect their bioactive efficiency. Furthermore, when dissolved within the polymeric matrix, these biomolecules may influence polymer foaming, especially at a high concentration. For these reasons, the choice of the processing conditions for scaffold bioactivation requires deep understanding of the interactions between the materials, the biomolecule, and the blowing agent. It is also desirable to develop novel strategies for the preparation and optimization of gradients of biomolecules within foamed scaffolds. In fact, these gradients are necessary to guide key tissue regeneration processes, namely cell migration and scaffold vascularization.

As shown in the recent literature reported in this review, the achievement of these goals can be helped by using high-pressure foaming in combination with CAD-based processes, and this field of research can open new avenues for the fabrication of multifunctional scaffolds for TE.

## Figures and Tables

**Figure 1 jfb-14-00480-f001:**
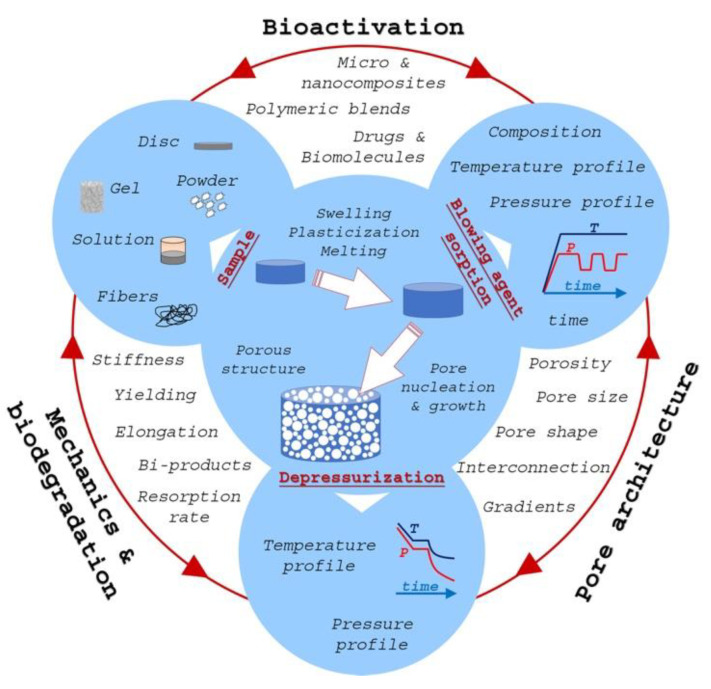
The different steps and implications in gas foaming of polymers for tissue engineering applications.

**Figure 2 jfb-14-00480-f002:**
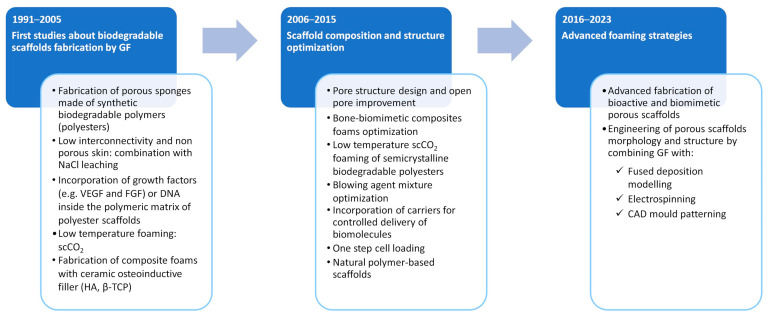
Review of the recent history and main challenges in gas foaming of scaffolds in tissue engineering.

**Figure 3 jfb-14-00480-f003:**
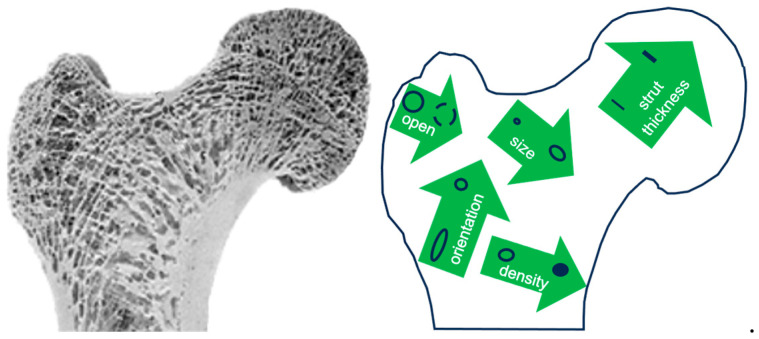
Details of the pore morphology in a human femur. Gradients in pore size, orientation, density, strut-to-wall ratio, and open-to-closed cell features are highlighted.

**Figure 4 jfb-14-00480-f004:**
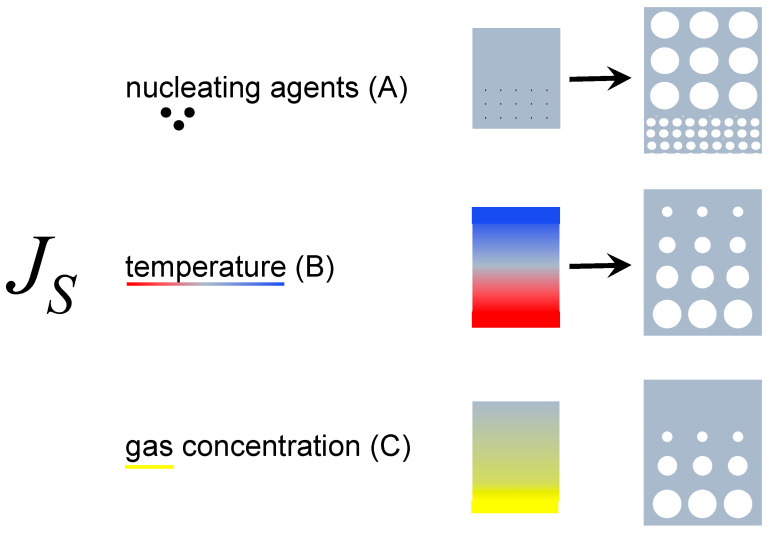
Available mechanisms to drive the nucleation rate, and *J_s_* to attain graded foams.

**Figure 5 jfb-14-00480-f005:**
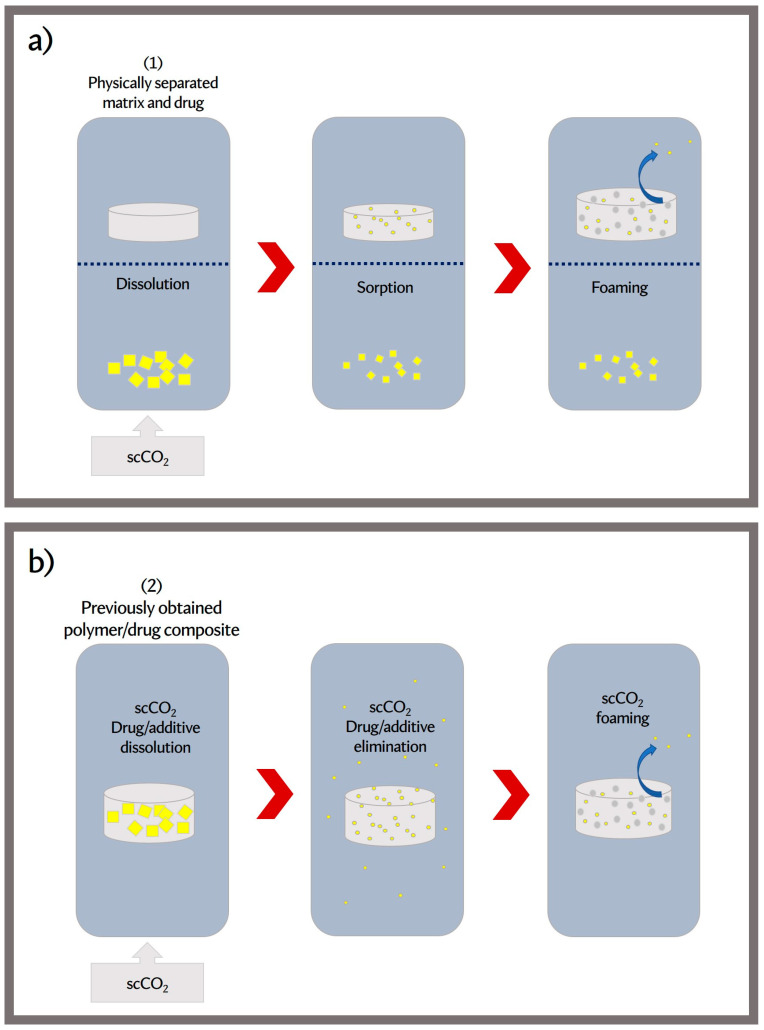
Drug impregnation procedures available with gas foaming. (**a**) Physically separated matrix and drug, (**b**) Previously prepared polymer/drug compound.

## Data Availability

Not applicable.

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
