# Peer review of "Current Trend and New Opportunities for Multifunctional Bio-Scaffold Fabrication via High-Pressure Foaming"

_jfb, 2023, doi:10.3390/jfb14090480_

Round 1
Reviewer 1 Report
The manuscript presents an engaging overview of the utilization of biocompatible and biodegradable foams created through the high-pressure foaming technique. This technique has garnered considerable attention in recent years as a promising avenue for generating porous scaffolds, both in vitro and in vivo, for tissue growth. The unique advantage of low-temperature operation during foaming is highlighted, facilitating the incorporation of bioactive molecules and cells within the scaffold's pores. Furthermore, the meticulous control over scaffold properties, such as density and pore architecture, through the careful selection of materials and processing conditions is commendably discussed. The significance of employing high-pressure foaming to curtail the use of cytotoxic and tissue-toxic compounds during scaffold fabrication is an important point. This aspect underlines the technique's potential for advancing scaffold biocompatibility.
The manuscript's objective to elucidate the current state-of-the-art and emerging trends in high-pressure foaming of biomedical polymers and composites for multifunctional tissue engineering scaffolds is commendable. The incorporation of gas foaming for bio-scaffold design and fabrication, as well as the emphasis on biomimetic porosity engineering, is well-articulated. Furthermore, the exploration of bioactive fillers and drugs for scaffold bioactivation adds depth to the discussion. Overall, this comprehensive review provides valuable insights into the realm of high-pressure foaming in tissue engineering. The manuscript is well-organized, informative, and could serve as a valuable resource for researchers in the field. The careful attention to current research trends and the potential to advance tissue engineering scaffold design make it a suitable candidate for publication. I recommend its acceptance for publication, with minor suggestions for improvement detailed below:
1. I recommend verifying the typesetting according to the journal's requirements, as the main text occupies only approximately four-fifths of the total page layout starting from page 2. However, both Figure 2 and Table 2 occupy the entire page layout.
2. The title of Table 2 is missing.
3. I recommend adding a conclusion at the end.
Author Response
We are grateful to the reviewer for his/her appraisal of the paper and for the constructive comments. Here a reply to the points raised.
- I recommend verifying the typesetting according to the journal's requirements, as the main text occupies only approximately four-fifths of the total page layout starting from page 2. However, both Figure 2 and Table 2 occupy the entire page layout.
- the manuscripts has been thoughtfully checked and numerous corrections have been made (see the marked version of the revised manuscript).
- The title of Table 2 is missing.
- we included the caption to Table 2.
- I recommend adding a conclusion at the end.
- we added the concluding section.
Reviewer 2 Report
This manuscript is a review of scaffold fabrication by gas foaming. It contains some useful information. However, it is likely that the manuscript was written in haste. Note the following:
1. There are many typo errors; need to be corrected.
2. Where are keywords?
3. No table caption in Table 2
4. The first paper was published in 1991. So, the year 1991 should be included in the period of Figure 1. Also, the year 2023 should be included. The sentences in section 3 are too long. It is preferable to group by attaching subheadings. For example, 3.1 1991-2005 period, 3.2 2006-2015 period, 3.3 2016-2023 period.
5. Long sentences should be broken into short paragraphs to make it easier for readers to read.
6. Unit spacing. All units must be spaced apart from the number, but % and ℃ must be attached to the number.
7. In Table 2, it is recommended to add a column for the target organs of each study.
8. At the end of the manuscript, it is recommended to add a section "Conclusions and recommendations for future works."
9. Contents of recent papers published in 2023 should be included more and cited in the manuscript.
10. The bibliography format is all messed up. All references must follow the journal's format guide.
Author Response
We thank the reviewer for his/her constructive comments. A reply to the point raised follows.
- There are many typo errors; need to be corrected.
The manuscript has been thoughtfully checked and numerous corrections have been made (see the marked version of the revised manuscript).
- Where are keywords?
Keywords were included in the amended version of the manuscript.
- No table caption in Table 2
Caption to Table 2 was added in the amended version of the manuscript.
- The first paper was published in 1991. So, the year 1991 should be included in the period of Figure 1. Also, the year 2023 should be included. The sentences in section 3 are too long. It is preferable to group by attaching subheadings. For example, 3.1 1991-2005 period, 3.2 2006-2015 period, 3.3 2016-2023 period.
Figure 1 was amended.
- Long sentences should be broken into short paragraphs to make it easier for readers to read.
Sentences were thoughtfully revised.
- Unit spacing. All units must be spaced apart from the number, but % and ℃ must be attached to the number.
Unit spacing was thoughtfully corrected.
- In Table 2, it is recommended to add a column for the target organs of each study.
We believe that the original version is more readable.
- At the end of the manuscript, it is recommended to add a section "Conclusions and recommendations for future works."
Conclusions are included in the amended version of the manuscript.
- Contents of recent papers published in 2023 should be included more and cited in the manuscript.
We include some pieces of the recent literature in the amended version of the manuscript.
- The bibliography format is all messed up. All references must follow the journal's format guide.
Bibliography style was amended.